# Effects of Copper Sulfate and Encapsulated Copper Addition on *In Vitro* Rumen Fermentation and Methane Production

Martyna Wilk [1,*], Ewa Pecka-Kiełb [2], Jerzy Pastuszak [3], Muhammad Umair Asghar [1] and Laura Mól [3]

1 Department of Animal Nutrition and Feed Science, Wrocław University of Environmental and Life Sciences, 51-630 Wrocław, Poland
2 Department of Animal Physiology and Biostructure, Wrocław University of Environmental and Life Sciences, 51-631 Wrocław, Poland
3 BioDose Sp. z o. o. Sp. k., 60-453 Poznań, Poland
* Correspondence: martyna.wilk@upwr.edu.pl

**Abstract:** Copper is a microelement crucial for the proper functioning of animals' metabolic processes. The function of copper in rumen fermentation processes and methanogenesis is not well analyzed. The aim of the study was to evaluate the different types of copper supplement, their rumen decomposition and effect on *in vitro* ruminal fermentation as well as methanogenesis. Two different copper additives were used in the experiment: CS—copper sulfate ($CuSO_4 \times 5\ H_2O$)—and EC—encapsulated copper (tribasic copper chloride and copper sulfate enclosed within a polysaccharide polymer coating). A total mixed ration without copper additive was used as a control (C). *In vitro* rumen fermentation was conducted, and fermentation profile, gas production and methanogenesis were evaluated. After 24 h of fermentation, the amount of copper in the rumen fluid was significantly higher in the CS group. EC was protected against rumen degradation to a greater extent. The type of used copper supplement affects rumen fermentation. However, the effect on methanogenesis is ambiguous. CS supplement increases rumen gas production but does not affect methanogenesis. The obtained results suggest that the EC supplement may reduce the risk of low-fat milk and may improve the economic indicators of milk production. An *in vivo* experiment is necessary to compare the obtained *in vitro* results with animal productivity.

**Keywords:** coated copper sulfate; rumen fermentation; methanogenesis; encapsulation

## 1. Introduction

Copper (Cu), as an essential trace element and is required for many enzymatic reactions involved in red blood cell and hormone production, as well as energy manufacturing [1]. Copper is involved in numerous bacterial activities, acting as a cofactor for important enzymes such as cytochrome oxidase, NADH dehydrogenase and superoxide dismutase [2]. Cu is essential for both ruminants and ruminal microorganisms as a redox protein cofactor or structural element [3–5]. It is worth emphasizing that excess copper can be extremely toxic. For example, a high level of Cu (100–200 µg/g, reported as tribasic copper chloride) had no effect on the steers' daily gain [6] or goats' growth performance improvement without toxicity signs [7]. However, in lambs or cows, the same concentration caused toxicities [3]. On the other hand, dietary supplementation with copper sulfate increased average daily gain, feed efficiency and the total tract digestibility of DM in bulls [8,9]. Lopez-Guisa and Satter [10] reported that the addition of Cu and Co above the NRC requirements may aid in the digestion of low-quality forage. The findings suggest that dietary copper has an invigorating effect on rumen nutrient digestion and microbial development. However, Gould and Kendall [11] reported that free copper ions from rumen-soluble copper sulfate might react with sulfur, molybdenum and iron to create insoluble complexes in the rumen, reducing the proportion of dietary copper absorption in the gut.

According to the NRC, the needs of ruminants for Cu vary depending on the species and amounts up to 8–10 μg/g [3,12]. The current widely used inorganic Cu supplement is copper sulfate; however, the rate of absorption of dietary Cu in mature dairy cows was just 15% [3]. Moreover, copper combined with a high molybdenum content (4–12 μg/mL of rumen fluid) also generates thiomolybdate, which reduces Cu availability [11]. Copper bioavailability in ruminants is limited because of interactions between Cu and other diet components that generate insoluble compounds that are not absorbed by the small intestinal tract. As a result, ruminants should be given a copper supplement with a lower rumen solubility than copper sulfate [13]. Wu et al. [14] reported that the solubility of copper sulfate is approximately 100%, but coated copper sulfate released 24.3% of copper in the rumen and 59.6% of copper in the gut. According to Wang et al. [15], the dietary supplementation of coated copper resulted in higher milk yield, apparent nutrient digestibility, rumen total VFA content and the quantity of fiber-digesting bacteria compared to those consuming CS-supplemented diets.

As mentioned above, copper is a crucial microelement in maintaining animal health and the efficient functioning of metabolic systems, but its specific role in microbial fermentation and its influence on methanogens in the rumen are poorly understood. Several methods have been investigated and implemented to minimize rumen methane generation, including appropriate nutritional practices, e.g., well-balanced diets, choice of proper feed materials and the use of enzymatic feed additives or biological inoculants that reduce the production of methane [16–18]. Selected yeast strains, essential oils, polyphenols, tannins, saponin and oils, which are rich in long-chain fatty acids, reduce methane release in ruminants that are mainly fed on roughage [19–22]. However, little is known about the exact effects of copper on ruminal physiology. As reported by Napasirth et al. [23], exceptionally high copper sulfate dosages (0.2–0.4%) in a ruminal *in vitro* culture suppress methane emissions.

The aim of the study was to evaluate the different types of copper supplements and their rumen decomposition, effect on *in vitro* ruminal fermentation processes, and methane generation.

## 2. Materials and Methods

### 2.1. Substrate

The total mixed ration (TMR) was formulated with whole-plant corn silage, haylage and silage beet pulp at a ratio of 740:130:130 g per kg of fresh matter [24]. Two different copper additives were used in the experiment: CS—copper sulfate ($CuSO_4 \times 5\,H_2O$)—and EC—encapsulated copper (CoProTex, BioDose, Poland; mix of tribasic copper chloride and copper sulfate placed inside a lipid matrix, enclosed within a polysaccharide polymer coating). A TMR without copper additive was used as a control (C). In both experimental groups, the copper compounds were added in an amount to ensure the supply of copper at the level of 12 mg/kg of fresh TMR.

### 2.2. Proximate Analysis

Proximate analysis was performed for the following variables: dry matter (DM, AOAC: 934.01); crude protein (CP, AOAC: 984.13) with the use of a FOSS Tecator 2300 Kjeltec Analyzer Unit (FOSS Tecator, Hoganas, AB, Sweden); crude ash (CA, AOAC: 942.05); ether extract (EE, AOAC: 920.39A) with the use of a BUCHI Extraction System B-811 (BÜCHI, Flawil, Switzerland); crude fiber (CF, AOAC: 978.10); fiber fractions, such as neutral detergent fiber (NDF) and acid detergent fiber (ADF, AOAC: 973.18), with the use of a ANKOM200 Fiber Analyzer (ANKOM Technology, Macedon, NY, USA) [25]. Acid detergent lignin (ADL) was determined according to the procedure of Van Soest et al. [26]. The obtained values of NDF, ADF and ADL were used to calculate hemicellulose, cellulose

and lignin. The nitrogen-free extractives (NFE) and the non-fiber carbohydrates (NFC) were calculated in g/kg of DM using the formulas:

$$NFE = 1000 - CP + CA + CF + EE),$$

$$NFC = 1000 - (CA + CP + EE + NDF),$$

where NFE—nitrogen-free extractives, CP—crude protein, CA—crude ash, CF—crude fibre, EE—ether extract, NFC—non-fibre carbohydrates and NDF—neutral detergent fiber.

The copper content of the total mixed ratio was determined by spectrophotometric method, which involved the use of a VARIAN AA240FS atomic absorption spectrometer (AA240FS, Varian, Palo Alto, CA, USA). The readings were taken at a wavelength of 324.8 nm for detecting copper (Cu; AOAC: 975.03) [25]. The chemical composition of the total mixed ration used in the present study is shown in Table 1.

**Table 1.** Chemical composition of the total mixed ration (mean $\pm$ sd).

| Chemical Composition | |
|---|---|
| Dry matter [g/kg] | $291.80 \pm 0.10$ |
| Crude ash [% DM] | $4.62 \pm 0.10$ |
| Crude protein [% DM] | $8.20 \pm 0.04$ |
| Ether extract [% DM] | $2.20 \pm 0.02$ |
| Crude fiber [% DM] | $20.04 \pm 0.03$ |
| NDF [% DM] | $51.17 \pm 0.02$ |
| ADF [% DM] | $25.28 \pm 0.07$ |
| ADL [% DM] | $3.44 \pm 0.01$ |
| Cellulose [% DM] | $21.84 \pm 0.06$ |
| Hemicellulose [% DM] | $25.89 \pm 0.05$ |
| Lignin [% DM] | $3.44 \pm 0.01$ |
| NFE [% DM] | $64.96 \pm 0.09$ |
| Cu [mg/kg DM] | $5.46 \pm 0.46$ |

*2.3. In Vitro Rumen Fermentation Profile*

Rumen fluid collected from three close-up dairy cows, Polish Holstein Friesian breed, were used to determine the *in vitro* fermentation profile. The animals were housed in an open cattle-housing system and fed with a total mixed ration feeding system. The ruminant feeding requirements were used to compost animal diet [24]. The rumen fluid was collected using the probe two hours before the morning total mixed ration feeding into a pre-warmed thermos and transported to the laboratory. Rumen fluid was blended under $CO_2$ for 30 s and strained through four layers of cheesecloth into a preheated flask. The samples of rumen fluid were pooled, mixed and used for analyses in nine replications per each group.

Samples for incubation were prepared according to McDougall [27]. An air dried diet (1 gram) and ruminal fluid (40 mL) diluted by McDougall buffer solution (1:3) were added to 250 mL serum bottles (Sigma-Aldrich, St. Louis, MO, USA), homogenized and flushed in $CO_2$. Fermentation was performed under anaerobic conditions in a water-bath shaker at 39 °C for 24 h. The gas formed during 24 h of fermentation was collected for analysis by using a gas-tight syringe. The methane content was determined by using a gas chromatograph with flame ionization detection and thermal conductivity detection (7890A GC, Agilent Technologies, Santa Clara, CA, US). The pH value of liquid samples was measured by a pH meter (CP-401; Elmetron, Poland). Liquid samples were centrifuged (15 min, 13,000 rpm), and formic acid (0.1 mL/2 mL of sample) was added to inhibit the fermentation processes. The VFA concentration and the percentage of individual acids:

acetic, propionic, isobutyric, butyric, isovaleric, valeric and isocaproic were determined using a gas chromatograph (7890A GC, Agilent Technologies, Santa Clara, CA, USA). The identification and concentration of individual acids in the analyzed samples was performed using ChemStation software (Agilent Technologies, Inc., Santa Clara, CA, USA)

The acid rations acetic to propionic (A:P) and propionic to butyric (P:B) were calculated. Fermentation efficiency (*FE*) was calculated by the Baran and Žitňan equation [28], and the efficiency of fermented hexose energy to *VFA* energy (*E*1) and methane energy (*E*2) was calculated according to Czerkawski [29] and IAEA [30].

$$FE = \frac{(0.622A + 1.092P + 1.56B + iB) \times 100}{(A + P + 2B)}$$

$$E1 = \frac{VFA\ energy}{fermented\ hexose\ energy} = \frac{62 + 0.47(P + 2B + 2V) \times 100}{(100 + B + V)}$$

$$E2 = \frac{methane\ energy}{fermented\ hexose\ energy} = \frac{28 - 0.47(P + V) \times 100}{(100 + B + V)}$$

where $A$, $P$, $B$, $V$ and $iB$ represent the proportions of acetic, propionic, butyric, valeric and isobutyric acids in the total *VFA* concentration, respectively.

The ratio of non-glucogenic to glucogenic *VFA*, known as the *VFA* utilization index (NGR), was expressed, according to equation:

$$NGR = \frac{A + 2B + V + iB + iV}{P + V + iV + iB}$$

where $A$, $P$, $B$ and $V$ represent the molar proportions of acetic acid, propionic acid, butyric acids and valeric acid, as well as branched chain fatty acids–isobutyric acid (*iB*) and isovaleric acid (*iV*) in the total *VFA* concentration [31].

Moreover, *in vitro* truly degraded dry matter (ivTDDM) was determined according to the procedure of Van Soest et al. [26].

### 2.4. Determination of Copper after In Vitro Rumen Fermentation

After 24 h of *in vitro* fermentation, rumen fluid was collected and strained through four layers of cheesecloth and freeze-dried using a laboratory freeze-dryer Alpha 1–4 LSCbasic (Christ, Germany). The copper content in the rumen fluid after *in vitro* fermentation was determined according to the method described above.

### 2.5. Statistical Analysis

All obtained numerical data was analyzed with the Shapiro–Wilk test. *In vitro* fermentation data for main effects (kind of plant material, addition of inoculum) were analyzed by two-way ANOVA using Statistica ver. 13.3 [32]. Additionally, obtained data for individual treatments was analyzed by one-way analysis of variance. The Duncan test was used to confirm significant differences between the means of the groups. Differences with $p \leq 0.05$ and $p \leq 0.01$ were considered significant and highly significant, respectively.

## 3. Results

After 24 h of *in vitro* fermentation, there were no significant differences in the pH value of rumen fluid and the value of ivTDDM between experimental groups (Table 2). Despite the same level of copper in the experimental doses, after 24 h of *in vitro* fermentation, the amount of copper in the rumen fluid was significantly higher in the CS group ($p = 0.0000$). The concentration of copper in the EC group was similar to the control, which proves that nearly 100% of the copper was protected against rumen decomposition.

**Table 2.** Effects of copper sulphate (CS) and encapsulated copper (EC) addition after 24 h of rumen fermentation (*in vitro*).

|  | Control | CS | EC | *p*-Value |
|---|---|---|---|---|
| pH initial (0 h) | 7.37 ± 0.06 | 7.36 ± 0.04 | 7.40 ± 0.07 | 0.4750 |
| pH final (24 h) | 6.43 ± 0.06 | 6.39 ± 0.07 | 6.45 ± 0.09 | 0.4770 |
| ivTDDM (%) | 58.91 ± 2.67 | 59.57 ± 2.40 | 57.59 ± 3.56 | 0.0885 |
| Cu content [mg/kg of dry rumen fluid] | 31.90 ± 4.21 [A] | 56.20 ± 6.69 [B] | 30.00 ± 5.01 [A] | 0.0000 |
| Cu content [mg/kg of rumen fluid] | 0.40 ± 0.05 [A] | 0.70 ± 0.08 [B] | 0.38 ± 0.06 [A] | 0.0000 |

[AB]—values in columns with different superscripts are significantly different at $p \leq 0.01$; mean value of copper additive: CS—11.76 ± 0.57; EC—12.24 ± 0.53.

Effects of copper sulfate and encapsulated copper addition on *in vitro* fermentation profile present Table 3. The use of encapsulated copper (EC) contributed to significantly ($p < 0.05$) lower gas production during the 24-hour period of *in vitro* fermentation, compared to CS. The type of used copper did not affect the production of methane and total VFA ($p > 0.05$). However, statistical analysis showed significant differences in propionate ($p < 0.01$) and heptanoate ($p < 0.05$) concentrations. Propionate was significantly higher, and heptanoate was significantly lower in the CS group compared to the EC and control groups. The type of used copper significantly affects the rumen fermentation indicators, such as A:P ratio, FE, E1, NGR ($p < 0.05$) and E2 ($p < 0.01$).

**Table 3.** Effects of copper sulfate (CS) and encapsulated copper (EC) addition on fermentation profile (*in vitro*).

|  | Control | CS | EC | *p*-Value |
|---|---|---|---|---|
| Gas production (mL) | 103.66 ± 10.13 [a] | 114.49 ± 6.17 [b] | 99.90 ± 9.19 [a] | 0.0279 |
| $CH_4$ (%) | 36.95 ± 1.71 | 33.96 ± 3.02 | 34.03 ± 3.58 | 0.1541 |
| Total VFA [x] | 223.90 ± 16.48 | 227.27 ± 20.09 | 221.15 ± 5.68 | 0.7899 |
| Total VFA [z] | 74.63 ± 5.49 | 75.76 ± 6.70 | 73.72 ± 1.89 | 0.2946 |
| Acetate [z] | 49.80 ± 5.01 | 48.13 ± 5.95 | 48.81 ± 1.36 | 0.8176 |
| Propionate [z] | 17.55 ± 0.39 [A] | 20.83 ± 2.55 [B] | 17.62 ± 1.33 [A] | 0.0056 |
| Isobutyrate [z] | 0.56 ± 0.08 | 0.53 ± 0.05 | 0.56 ± 0.09 | 0.7120 |
| Butyrate [z] | 5.45 ± 0.48 | 4.97 ± 0.95 | 5.41 ± 1.08 | 0.5912 |
| Isovalerate [z] | 0.40 ± 0.04 | 0.42 ± 0.06 | 0.42 ± 0.06 | 0.7452 |
| Valerate [z] | 0.42 ± 0.05 | 0.44 ± 0.06 | 0.44 ± 0.09 | 0.8406 |
| Isocaproate [z] | 0.21 ± 0.01 | 0.20 ± 0.01 | 0.21 ± 0.01 | 0.3239 |
| Hexanoate [z] | 0.16 ± 0.01 | 0.16 ± 0.03 | 0.17 ± 0.02 | 0.6795 |
| Heptanoate [z] | 0.09 ± 0.00 [a] | 0.08 ± 0.00 [b] | 0.09 ± 0.01 [a] | 0.0180 |
| A:P | 2.84 ± 0.27 [a] | 2.34 ± 0.41 [b] | 2.78 ± 0.25 [a] | 0.0312 |
| P:B | 3.24 ± 0.27 | 4.44 ± 1.61 | 3.38 ± 0.74 | 0.1236 |
| FE [%] | 75.70 ± 0.77 [a] | 77.32 ± 1.51 [b] | 75.86 ± 0.74 [a] | 0.0351 |
| E1 [%] | 75.00 ± 0.43 [a] | 76.11 ± 0.98 [b] | 75.01 ± 0.99 [a] | 0.0458 |

**Table 3.** *Cont.*

|  | Control | CS | EC | *p*-Value |
|---|---|---|---|---|
| E2 [%] | 20.02 ± 0.17 [A] | 18.51 ± 1.18 [B] | 19.98 ± 0.57 [A] | 0.0052 |
| NGR | 3.28 ± 0.28 [a] | 2.71 ± 0.47 [b] | 3.22 ± 0.27 [a] | 0.0256 |

[x] mmol/L of undiluted ruminal fluid; [z] mol/100 mol of total VFA concentration; [ab, AB]—values in columns with different superscripts are significantly different at $p \leq 0.05$ and $p \leq 0.01$, respectively; A:P—acetate:propionate ratio; P:B—propionate:butyrate ratio; NGR—the VFA utilization index; FE—fermentation efficiency; E1—efficiency of fermented hexose energy to VFA energy; E2—efficiency of fermented hexose energy to methane energy; mean value of copper additive: CS—11.76 ± 0.57; EC—12.24 ± 0.53.

## 4. Discussion

It is known that the initial pH affects some free Cu ions in a medium and has a significant impact on bacteria's capacity to proliferate [33]. However, initial rumen fluid in all experimental groups stayed within the optimal range (6.5 to 7.5) and did not show significant differences [34]. Despite differences in fatty acid profiles noted during *in vitro* fermentation, the pH value of rumen fluid was similar in all experimental groups, without statistical differences. Similarly, Engle and Spears et al. [35] reported that supplementation of 10–20 g of Cu in the culture medium had no effect on pH. On the other hand, Hasman et al. [33] reported that 40 g of Cu influenced the pH of rumen fluid. The pH levels measured in the presented study stayed within the optimal range (6.14–6.45) for bacterial metabolism and proliferation [36].

After 24 h of *in vitro* fermentation, the amount of copper in the rumen fluid was significantly higher in the CS group (56.20 mg/kg DM) compared to EC (30.00 mg/kg DM) and control (31.90 mg/kg DM). This result allows us to conclude that nearly 100% of the analyzed supplement of encapsulated copper (EC) was protected against rumen decomposition.

The gas production results suggest that CS supplement improves the degradability of the substrate. However, despite significant differences in the amount of producing gas, the level of ivTDDM was similar in all experimental groups (approximately 58–60%). The use of encapsulated copper (EC) contributed to a significant ($p < 0.05$) reduction of gas production during the 24 h period of *in vitro* fermentation compared to the CS supplementation, which can be related to the varying levels of volatile fatty acid production. On the other hand, the type of copper supplement has no effect on methane generation. Vázquez et al. [37] reported that bacterial growth is enhanced when the concentration of Cu in the growth medium is less than 0.5 mg/kg of DM. However, when the concentration of the mineral reaches this threshold, bacterial proteolysis occurs [37]. Hernández-Sánchez et al. [38] reported that raising the amount of copper sulfate from 20, 40 to 60 mg/kg DM had no effect on VFA concentration on *in vitro* trials. On the other hand, increasing the concentration of copper sulfate to 80 and 100 mg/kg DM decreased the total amount of rumen bacteria and, therefore, the concentration level of VFA. In the presented study, significant differences in the concentration of propionate and heptanoate between analyzed copper supplements were noted. However, the change in the concentration of these two acids did not affect the total VFA content in the analyzed samples, which is consistent with the results presented by Hernández-Sánchez et al. [38]. Wang et al. [15] reported that the total rumen content of VFA increases as the level of coated copper supplementation increases, which suggests that the encapsulated copper may promote the breakdown of nutrients in the rumen. Contrary, Vázquez-Armijo et al. [37] found that the addition of 21.7 mg/kg DM of copper sulfate improved ruminal short-chain fatty acid synthesis, gas generation and DM degradability *in vitro.* Similarly, studies with goats reported that the dietary addition of copper sulfate at 10, 20 and 30 mg/kg DM to feed improved ruminal concentrations of VFA, acetate, apparent total tract digestibility of DM, organic matter and fiber [39,40].

The concentration of propionate during the fermentation of CS samples was substantially ($p < 0.05$) higher compared to the EC and control. As reported by Shan et al. [9], the decrease in ruminal propionate percentage corresponds with a reduction of amylase activity and concentrations of *P. ruminicola* and *Rb. amylophilus.* Consequently, a lower

propionate molar percentage can reduce hepatic glucose synthesis, resulting in a drop in milk lactose output and blood glucose levels in dairy cows [41]. The hydrogenesis in the gastrointestinal tract is determined by VFA profile. For example, metabolic hydrogen is captured by propionate, thereby reducing methane production from unit-fermented organic mass [42,43]. However, despite the differences in the propionate level between the analyzed samples, the differences in methane production were not confirmed. Moreover, the stimulation of rumen acetogenic microbes, consuming $H_2$ to form acetate, potentially reduces $CH_4$ production. Yue et al. [44] reported that the decrease in $CH_4$ formation was related to a decrease in H2 with high Cu levels. Zhang et al. [36] found that adding 10–20 g Cu/g DM reduced acetate without changing propionate and butyrate concentrations; this might be because the majority of the acetate had really been transformed to $CH_4$ [16]. Napasirth et al. [23] conducted an investigation in which they assessed dosages of 0.2% and 0.4% $CuSO_4$ in a ruminal *in vitro* propagation and reported that copper sulfate is a viable way of reducing $CH_4$ emissions. However, these quantities are exceedingly high, exceed the needed amounts and may induce ruminant poisoning [3,12]. In the current investigation, there were no statistical differences in acetate and $CH_4$ generation between the analyzed types of copper supplement.

The differences in acetate:propionate ratio between samples were noted ($p < 0.05$). The ratio of non-glucogenic VFA to glucogenic VFA is associated with methanogenesis and therefore energy balance and milk composition [45]. Propionate, which is glucogenic fatty acid, is responsible for the deposition of energy in the body tissues. Which is why the low value of NGR (<3.0) increases the risk of low-fat milk. The higher level of rumen propionate production in the CS group was connected with a statistically significant lower value of the NGR index. The NGR index in the group with the EC supplement addition was at the level of 3.22, which shows that this form of copper may reduce the risk of producing milk with low fat content. Moreover, low NGR is connected with lower losses of energy in the form of fermentation gasses [31]. However, the differences in methane production were not confirmed in this study. The E2 index, the efficiency of fermented hexose energy to methane energy, was significantly lower ($p < 0.01$) in the CS group compared to ES or control. Similarly to the NGR index, it was not reflected in the results of methanogenesis.

The FE index evaluates rumen fermentation as an effect of feed additives and microbial metabolism modulation. Both indexes, FE and E1 (efficiency of fermented hexose energy to VFA energy index), were statistically higher ($p < 0.05$) in the CS group compared to EC and control. However, these differences were not high.

## 5. Conclusions

According to this study, the type of copper supplement used in the diet might affect rumen fermentation. The effect of the type of copper addition on the level of methanogenesis is ambiguous. In the presented study, noted differences in the rumen fermentation indexes indicate a beneficial effect of the addition of CS on the reduction of methanogenesis. On the other hand, CS supplement increases rumen gas production but does not affect methanogenesis. On the contrary, EC supplement improved the NGR index, which shows that this form of copper reduces the risk of low-fat milk and will improve the economic indicators of milk production. A continuation of the presented *in vitro* experiment *in vivo* with dairy cows is planned to assess the impact of the analyzed copper supplements on methane emissions, rumen fermentation and animal productivity.

**Author Contributions:** Conceptualization, M.W. and J.P.; methodology, M.W., J.P. and E.P.-K.; software, M.W.; validation, M.W.; formal analysis, M.W., J.P. and E.P.-K.; investigation, M.W. and E.P.-K.; resources, M.W., J.P. and E.P.-K.; data curation, M.W., E.P.-K.; writing—original draft preparation, M.W., M.U.A. and L.M.; writing—review and editing, M.W. All authors have read and agreed to the published version of the manuscript.

**Funding:** The APC was funded by Wroclaw University of Environmental and Life Sciences, Poland.

**Institutional Review Board Statement:** Not applicable.

**Data Availability Statement:** Not applicable.

**Conflicts of Interest:** The authors declare no conflict of interest.

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
