# Peer review of "Effects of Copper Sulfate and Encapsulated Copper Addition on In Vitro Rumen Fermentation and Methane Production"

_agriculture, doi:10.3390/agriculture12111943_

Round 1
Reviewer 1 Report
The document entitled "Effects of copper sulfate and encapsulated copper addition on in vitro rumen fermentation and methane production" does have an important idea in reduction of methane from rumen fermentation. However, it is unclear of the mechanisms behind the use of copper in this reduction. It was even stated that copper had no effect on methane production, which was not surprising. In the abstract some statements were made "EC supplement reduces the risk of low-fat milk and may improve the economic indicators of milk production." My question is how can this conclusion be made and this experiment did not evaluate these parameters. These statements can be made when animal studies are conducted in vivo. This statement should be revised
The other major concern is that previous work done in vitro found that copper sulphate reduces methane emission at very high concentration which may have been due to destruction of specific microflora in the rumen as stated in the introduction " As reported by Napasirth et al. [23] exceptionally high copper sulfate dosages (0.2-0.4%) in a ruminal in vitro culture suppress methane emissions." If this was known it may have been more worthwhile to conduct an in vivo experiment to determine the effect of Cu on animal performance and ruminal physiology.
My final concern is the exact process in which copper will affect microbial fermentation, methane production and gas production. This was not addressed in the discussion or even introduction of this manuscript. This research should have been done with ruminal samples in vivo to quantify the species of microorganisms present when fed Cu in different forms.
Finally, there are some edits to be made with typos. See abstract "However" in the abstract the 'r' is missing. Please re read the entire manuscript to make revisions.
Author Response
Dear reviewer,
The authors would like to thank the reviewers for all comments and suggestions which may improve manuscript quality. The authors made changes to the text in line with the reviewers' renewed comments.
Thank you for all comments that have contributed to the improvement of the quality of the manuscript. Thank you for your comment, authors apologize for the unfortunately worded statement. The sentence “EC supplement reduces the risk of low-fat milk and may improve the economic indicators of milk production.” has been corrected.
The authors agree with the reviewer's opinion that it may have been more worthwhile to conduct an in vivo experiment to determine the effect of Cu on animal performance and ruminal physiology. However, the authors would like to emphasize that the main aim of the study was to compare the effects of copper sulfate and the new encapsulated copper supplement on rumen physiology which were administered at a safe level that could be used during in vivo trials. Due to the trend of scientific research towards improving the welfare of experimental animals, the standard procedure is to start testing the preparation excluding live animals. Where possible, in vitro analyzes should be used. The manuscript presents the results of a pilot study of both copper supplements on in vitro analyzes. An in vivo experiment which determines the influence of the analyzed preparations on the production indicators of cows is planned.
The authors hope that the corrections made in line with the reviewers' suggestions have improved the readability and quality of the manuscript.
Yours faithfully,
Authors
Reviewer 2 Report
This study evaluated two different types of copper supplementation on in vitro rumen fermentation and methane production. The experimental concept is clearly presented. However, there are some issues should discuss with authors.
1. The manuscript should be labelled line number.
2. Author may make a mistake with the crude ash content of the TMR in Table 1, because the number was so high, whose unit should be g/kg.
3. If the amount of substrate using in the fermentation was fresh matter basis or air dried basis? And what was the amount of fermentation inoculum used in the in vitro experiment?
4. How was the Cu content of rumen fluid determined? Determination of Cu content of dry rumen fluid was described, is that mean the dry matter content of rumen fluid was determined?
5. The first sentence of the paragraph above Table 3 was inappropriate because EC did not significant decrease gas production compared to control group.
6. What was the meaning of superscripts “x” in Table 3? And what was the difference between the two total VFA values?
7. Discussion of copper released in the rumen only by the content in the dry rumen fluid was not sufficient. Because content of copper in the rumen fluid was consisted not only with supplementation but also decomposition copper in the diet. From the gas production results we could conclude that CS improve the digestibility of the substrate, because the gas production was positive correlated with fermentable matter in the substrate, which meant more copper should release from the diet. Moreover, authors should determine copper content in both liquid and solid phase of the fermentation system so that we compared the bioavailability the supplementation because copper could not escape from the in vitro system.
8. It was reported by studies that production of propionate was H2 sink in the rumen, which would compete with CO2 to form CH4. Although in this study propionate was increased by CS, the methane production was not affected. Reason for that may because of the improved digestibility of the substrate, which provided more H2 both for CO2 reduction and propionate formation.
Author Response
Dear reviewer,
The authors would like to thank the reviewers for all comments and suggestions which may improve manuscript quality. The authors made changes to the text in line with the reviewers' renewed comments.
Thank you for all comments that have contributed to the improvement of the quality of the manuscript.
[question 1 and 2]
The crude ash content of the TMR in Table 1 was corrected and the line numbers were added.
[question 3]
Moreover, the missing information has been added to the text. The dry matter content of rumen fluid was determined by using freeze-dryer Alpha 1-4 LSCbasic (Christ, Germany).
[question 4]
The method of determining copper in rumen fluid was described in chapter 2.4. Determination of copper after in vitro rumen fermentation
“After 24 hours of in vitro fermentation, rumen fluid was collected and strained through four layers of cheesecloth and freeze-dried using a laboratory freeze-dryer Alpha 1-4 LSCbasic (Christ, Germany). The copper content in the rumen fluid after in vitro fermentation was determined according to the method described above.”
[question 5]
The sentence was corrected.
[question 6]
The explanation of superscripts “x” in Table 3 was added? Total VFA values are presented in two forms: x - mmol/L of undiluted ruminal fluid; z - mol/100 mol of total VFA concentration.
[question 7 and 8]
Thank you for this comment. The authors agree with the reviewer that the results of copper concentration in the solid phase would greatly enrich the discussion of copper released in the rumen. The digestibility of the substrate was determined in the experiment. IVTDDM did not differ significantly between trials (p = 0.0885). The data has been added to table 2, the authors hope that the added results will make the whole work more attractive.
In this study propionate was increased by CS, however the methane production was not affected, similarly ivTDDM. Determination of CO2 and N2 production would probably provide interesting data for discussion, however these parameters were not marked.
The authors hope that the corrections made in line with the reviewers' suggestions have improved the readability and quality of the manuscript.
Yours faithfully,
Authors